**Accelerated hydrological cycle over the Sanjiangyuan region induces**
**more streamflow extremes at different global warming levels**
Peng Ji[1,2], Xing Yuan[3*], Feng Ma[3], Ming Pan[4]
[1]Key Laboratory of Regional Climate-Environment for Temperate East Asia, Institute
of Atmospheric Physics, Chinese Academy of Sciences, Beijing 100029, China
[2]College of Earth and Planetary Sciences, University of Chinese Academy of Sciences,
Beijing 1000493, China
[3]School of Hydrology and Water Resources, Nanjing University of Information
Science and Technology, Nanjing 210044, China
[4]Department of Civil and Environmental Engineering, Princeton University, Princeton,
New Jersey, USA
*Correspondence to*: Xing Yuan (xyuan@nuist.edu.cn)

**Abstract.** Serving source water for the Yellow, Yangtze and Lancang-Mekong rivers, the Sanjiangyuan region concerns 700 million people over its downstream areas. Recent research suggests that the Sanjiangyuan region will become wetter in a warming future, but future changes of streamflow extremes remain unclear due to the complex hydrological processes over high-land areas and limited knowledge of the influences of land cover change and $CO_2$ physiological forcing. Based on high resolution land surface modeling during 1979~2100 driven by the climate and ecological projections from 11 newly released Coupled Model Intercomparison Project Phase 6 (CMIP6) climate models, we show that different accelerating rates of precipitation and evapotranspiration at 1.5°C global warming level induce 55% more dry extremes over Yellow river and 138% more wet extremes over Yangtze river headwaters compared with the reference period (1985~2014). An additional 0.5°C warming leads to a further nonlinear and more significant increase for both dry extremes over Yellow river (22%) and wet extremes over Yangtze river (64%). The combined role of $CO_2$ physiological forcing and vegetation greening, which used to be neglected in hydrological projections, is found to alleviate dry extremes at 1.5 and 2.0°C warming levels but to intensify dry extremes at 3.0°C warming level. Moreover, vegetation greening contributes half of the differences between 1.5 and 3.0°C warming levels. This study emphasizes the importance of ecological processes in determining future changes in streamflow extremes, and suggests a "dry gets drier, wet gets wetter" condition over the warming headwaters.

**Keywords** Terrestrial hydrological cycle, streamflow extremes, global warming levels,

38 CMIP6, Sanjiangyuan, land cover change

## 1 Introduction

Global temperature has increased at a rate of 0.17°C/decade since 1970, contrary to the cooling trend over the past 8000 years (Marcott et al., 2013). The temperature measurements suggest that 2015-2019 is the warmest five years and 2010-2019 is also the warmest decade since 1850 (WMO, 2020). To mitigate the impact of this unprecedented warming on the global environment and human society, 195 nations adopted the Paris Agreement which decides to "hold the increase in the global average temperature to well below 2°C above pre-industrial levels and pursing efforts to limit the temperature increase to 1.5°C".

The response of regional and global terrestrial hydrological processes (e.g., streamflow and its extremes) to different global warming levels has been investigated by numerous studies in recent years (Döll et al., 2018; Hoegh-Guldberg et al., 2018; Marx et al., 2018; Mohammed et al., 2017; Thober et al., 2018; Xu et al., 2019; Zhang et al., 2016). In addition to climate change, recent works reveal the importance of the ecological factors (e.g., the $CO_2$ physiological forcing and land cover change), which are often unaccounted for in hydrological modeling works, in modulating the streamflow and its extremes. For example, the increasing $CO_2$ concentration is found to alleviate the decreasing trend of streamflow in the future at global scale through decreasing the stomatal conductance and vegetation transpiration (known as the $CO_2$ physiological forcing) (Fowler et al., 2019; Wiltshire et al., 2013; Yang et al., 2019; Zhu et al., 2012). Contrary to the $CO_2$ physiological forcing, the vegetation greening in a warming climate is found to play a significant role in exacerbating hydrological

drought, as it enhances transpiration and dries up the land (Yuan et al., 2018b).
However, the relative contributions of $CO_2$ physiological forcing and vegetation
greening to the changes in terrestrial hydrology especially the streamflow extremes
are still unknown, and whether their combined impact differs among different
warming levels needs to be investigated.
Hosting the headwaters of the Yellow river, the Yangtze river and the
Lancang-Mekong river, the Sanjiangyuan region is known as the "Asian Water
Tower" and concerns 700 million people over its downstream areas. Changes of
streamflow and its extremes over the Sanjiangyuan region not only influence the local
ecosystems, environment and water resources, but also the security of food, energy,
and water over the downstream areas. Both the regional climate and ecosystems show
significant changes over the Sanjiangyuan region due to global warming (Bibi et al.,
2018; Kuang and Jiao, 2016; Liang et al., 2013; Yang et al., 2013; Zhu et al., 2016).
Historical changes of climate and ecology (e.g. land cover) are found to cause
significant reduction in mean and high flows over the Yellow River headwaters during
1979-2005, which potentially increases drought risk over its downstream areas (Ji and
Yuan, 2018). And the $CO_2$ physiological forcing is revealed to cause equally large
changes in regional flood extremes as the precipitation over the Yangtze and Mekong
rivers (Fowler et al., 2019). Thus the Sanjiangyuan region is a sound region to
investigate the role of climate change and ecological change (e.g., land cover change
and $CO_2$ physiological forcing) in influencing the streamflow and its extremes (Cuo et
al., 2014; Ji and Yuan, 2018; Zhu et al., 2013). Recent research suggests that the
Sanjiangyuan region will become warmer and wetter in the future, and extreme
precipitation will also increase at the 1.5°C global warming level and further intensify
with a 0.5°C additional warming (Li et al., 2018; Zhao et al., 2019). However, how
the streamflow extremes would respond to the 1.5°C warming, what an additional
0.5°C or even greater warming would cause, and how much contributions do the
ecological factors (e.g., $CO_2$ physiological forcing and land cover change) have, are
still unknown. Solving the above issues is essential for assessing the climate and
ecological impact on this vital headwaters region.
In this study, we investigated the future changes in the streamflow extremes over
the Sanjiangyuan region from an integrated eco-hydrological perspective by taking
$CO_2$ physiological forcing and land cover change into consideration. The combined
impacts of the above two ecological factors at different global warming levels were
also quantified and compared with the impact of climate change. The results will help
understand the role of ecological factors in future terrestrial hydrological changes
over the headwater regions like the Sanjiangyuan, and provide guidance and support
for the stakeholders to make relevant decisions and plans.
**2 Data and methods**
**2.1 Study domain and observational data**
The Sanjiangyuan region is located at the eastern part of the Tibetan Plateau
(Figure 1a), with the total area and mean elevation being $3.61 \times 10^5$ km$^2$ and 5000 m
respectively. It plays a critical role in providing freshwater, by contributing 35%, 20%
and 8% to the total annual streamflow of the Yellow, Yangtze and Lancang-Mekong
rivers (Li et al., 2017; Liang et al., 2013). The source regions of Yellow, Yangtze and
Lancang-Mekong rivers account for 46%, 44% and 10% of the total area of the
Sanjiangyuan individually, and the Yellow river source region has a warmer climate
and sparser snow cover than the Yangtze river source region.

Monthly streamflow observations from the Tangnaihai (TNH) and the Zhimenda

(ZMD) hydrological stations (Figure 1a), which were provided by the local authorities,
were used to evaluate the streamflow simulations. Data periods are 1979-2011 and
1980-2008 for the Tangnaihai and Zhimenda stations individually. Estimations of
monthly terrestrial water storage change and its uncertainty during 2003-2014 were
provided by the Jet Propulsion Laboratory (JPL), which used the mass concentration
blocks (mascons) basis functions to fit the Gravity Recovery and Climate Experiment
(GRACE) satellite's inter-satellite ranging observations (Watkins et al., 2015). The
Model Tree Ensemble evapotranspiration (MTE_ET; Jung et al., 2009) and the Global
Land Evaporation Amsterdam Model evapotranspiration (GLEAM_ET) version 3.3a
(Martens et al., 2017) were used to evaluate the ET simulation.
**2.2 CMIP6 Data**

Here, 19 Coupled Model Intercomparison Project phase 6 (CMIP6, Eyring et al.,

2016) models which provide precipitation, near-surface temperature, specific
humidity, 10-m wind speed, surface downward shortwave and longwave radiations at
daily timescale were first selected for evaluation. Then, models were chosen for the
analysis when the simulated meteorological forcings (e.g., precipitation, temperature,
humidity, and shortwave radiation) averaged over the Sanjiangyuan region have the
same trend signs as the observations during 1979-2014. Table 1 shows the 11 CMIP6
models that were finally chosen in this study. For the future projection (2015-2100),
we chose two Shared Socioeconomic Pathways (SSP) experiments: SSP585 and
SSP245. SSP585 combines the fossil-fueled development socioeconomic pathway
and $8.5W/m^2$ forcing pathway (RCP8.5), while SSP245 combines the moderate
development socioeconomic pathway and $4.5 \ W/m^2$ forcing pathway (RCP4.5)
(O'Neill et al., 2016). Land cover change is quantified by leaf area index (LAI) as
there is no significant transition between different vegetation types (not shown)
according to the Land-use Harmonization 2 (LUH2) dataset
(https://esgf-node.llnl.gov/search/input4mips/). For the CNRM-CM6-1, FGOALS-g3
and CESM2, the ensemble mean of LAI simulations from the other 8 CMIP6 models
was used because CNRM-CM6-1 and FGOALS-g3 do not provide dynamic LAI
while the CESM2 simulates an abnormally large LAI over the Sanjiangyuan region.
To avoid systematic bias in meteorological forcing, the trend-preserved bias
correction method suggested by ISI-MIP (Hempel et al., 2013), was applied to the
CMIP6 model simulations at monthly scale. The China Meteorological Forcing
Dataset (CMFD) was taken as meteorological observation (He et al., 2020). For each
month, temperature bias in CMIP6 simulations during 1979-2014 was directly
deducted. Future temperature simulations in SSP245 and SSP585 experiments were
also adjusted according to the historical bias. Other variables were corrected by using
a multiplicative factor, which was calculated by using observations to divide
simulation during 1979-2014. In addition, monthly leaf area index was also adjusted
to be consistent with satellite observation using the same method as temperature. All
variables were first interpolated to the 10 km resolution over the Sanjiangyuan region
and the bias correction was performed for each CMIP6 model at each grid. After bias
correction, absolute changes of temperature and leaf area index, and relative changes
of other variables were preserved at monthly time scale (Hempel et al., 2013). Then,
the adjusted CMIP6 daily meteorological forcings were disaggregated into hourly
using the diurnal cycle ratios from the China Meteorological Forcing Dataset.
The historical $CO_2$ concentration used here is the same as the CMIP6 historical
experiment (Meinshausen et al., 2017), while future $CO_2$ concentration in SSP245 and
SSP585 scenarios came from simulations of a reduced-complexity carbon-cycle
model MAGICC7.0 (Meinshausen et al., 2020).
**2.3 Experimental design**
The land surface model used in this study is the Conjunctive Surface-Subsurface
Process model version 2 (CSSPv2), which has been proved to simulate the energy and
water processes over the Sanjiangyuan region well (Yuan et al., 2018a). Figure 2
shows the structure and main ecohydrological processes in CSSPv2. The CSSPv2 is
rooted in the Common Land Model (CoLM; Dai et al., 2003) with some
improvements at hydrological processes. CSSPv2 has a volume-averaged soil
moisture transport (VAST) model, which solves the quasi-three dimensional
transportation of the soil water and explicitly considers the variability of moisture flux
due to subgrid topographic variations (Choi et al., 2007). Moreover, the Variable
Infiltration Capacity runoff scheme (Liang et al., 1994), and the hydrological
properties of soil organic matters were incorporated into the CSSPv2 by Yuan et al.
(2018a), to improve its performance in simulating the terrestrial hydrology over the
Sanjiangyuan region. Similar to CoLM and Community Land Model (Oleson et al.,
2013), vegetation transpiration in CSSPv2 is based on Monin-Obukhov similarity
theory, and the transpiration rate is constrained by leaf boundary layer and stomatal
conductances. Parameterization of the stomatal conductance ( $g_s$ ) in CSSPv2 is
$$g_s = m \frac{A_n}{P_{CO_2}/P_{atm}} h_s + b\beta_t \qquad (1)$$

where the $m$ is a plant functional type dependent parameter, $A_n$ is leaf net
photosynthesis ( $\mu\,mol\,CO_2\,m^{-2}\,s^{-1}$ ), $P_{CO_2}$ is the CO$_2$ partial pressure at the leaf
surface ( $Pa$ ), $P_{atm}$ is the atmospheric pressure ( $Pa$ ), $h_s$ is the lead surface
humidity, $b$ is the minimum stomatal conductance ( $\mu\,mol\,m^{-2}\,s^{-1}$ ), while $\beta_t$ is the
soil water stress function. Generally, the stomatal conductance decreases with the
increasing of CO$_2$ concentration.
First, bias-corrected meteorological forcings from CMIP6 historical experiment
were used to drive the CSSPv2 model (CMIP6_His/CSSPv2). All simulations were
conducted for two cycles during 1979-2014 at half-hourly time step and 10 km spatial
resolution, with the first cycle serving as the spin-up. Correlation coefficient (CC) and
root mean squared error (RMSE) were calculated for validating the simulated monthly
streamflow, annual evapotranspiration and monthly terrestrial water storage. The
King-Gupta efficiency (KGE; Gupta et al., 2009), which is widely used in streamflow
evaluations, was also calculated. Above metrics were calculated as follows:
$$CC = \frac{\sum_{i=1}^{n}(x_i - \bar{x})(y_i - \bar{y})}{\sqrt{\sum_{i=1}^{n}(x_i - \bar{x})^2 \sum_{i=1}^{n}(y_i - \bar{y})^2}} \qquad (2)$$

$$RMSE = \sqrt{\frac{\sum_{i=1}^{n}(x_i - y_i)^2}{n}} \qquad (3)$$

$$KGE = 1 - \sqrt{(1-CC)^2 + (1-\frac{\sigma_x}{\sigma_y})^2 + (1-\frac{\bar{x}}{\bar{y}})^2} \qquad (4)$$

where $x_i$ and $y_i$ are observed and simulated variables in a specific month/year $i$
individually, and $\bar{x}$ and $\bar{y}$ are the corresponding monthly/annual means during the
evaluation period $n$. The $\sigma_x$ and $\sigma_y$ are standard deviations for observed and
simulated variables respectively. The correlation coefficient represents the correlation
between simulation and observation, while RMSE means simulated error. The KGE
ranges from negative infinity to 1, and model simulations can be regard as satisfactory
when the KGE is larger than 0.5 (Moriasi et al., 2007).
Second, bias-corrected meteorological forcings in SSP245 and SSP585 were
used to drive CSSPv2 during 2015-2100 with dynamic LAI and $CO_2$ concentration
(CMIP6_SSP/CSSPv2). Initial conditions of CMIP6_SSP/CSSPv2 came from the last
year in CMIP6_His/CSSPv2.
Then, the second step was repeated twice by fixing the monthly LAI
(CMIP6_SSP/CSSPv2_FixLAI)     and     mean     $CO_2$     concentration
(CMIP6_SSP/CSSPv2_FixCO2)     at     2014     level.     The     difference     between
CMIP6_SSP/CSSPv2 and CMIP6_SSP/CSSPv2_FixLAI is regarded as the net effect
of land cover change, and the difference between CMIP6_SSP/CSSPv2 and
CMIP6_SSP/CSSPv2_FixCO2 is regarded as the net effect of $CO_2$ physiological
forcing.
**2.4 Warming level determination**
A widely used time-sampling method was adopted to determine the periods of
different global warming levels (Chen et al., 2017; Döll et al., 2018; Marx et al., 2018;
Mohammed et al., 2017; Thober et al., 2018). According to the HadCRUT4 dataset
(Morice et al., 2012), the global mean surface temperature has increased by 0.66°C
from the pre-industrial era (1850-1900) to the reference period defined as 1985-2014.
Then, starting from 2015, 30-years running mean global temperatures were compared
to those of the 1985-2014 period for each GCM simulation. And the
1.5°C/2.0°C/3.0°C warming period is defined as the 30-years period when the
0.84°C/1.34°C/2.34°C global warming, compared with the reference period
(1985-2014), is first reached. The median years of identified 30-year periods, referred
as "crossing years", are shown in Table 2.
**2.5 Definition of dry and wet extremes and robustness assessment**
In this research, the standardized streamflow index (SSI) was used to define dry
and wet extremes (Vicente-Serrano et al., 2012; Yuan et al., 2017). The
July-August-September (JAS) mean streamflow for each year of the reference period
was first collected and used to fit a gamma distribution:
$$f(x,\beta,\alpha) = \frac{\beta^{\alpha}}{\Gamma(\alpha)} x^{\alpha-1} e^{-\beta x} \qquad (5)$$
where $x$ means streamflow, while $\alpha$ and $\beta$ are parameters. Then the fitted
distribution was used to standardize the JAS mean streamflow in each year ($i$) during

both the reference and projection periods as:

$$SSI_i = Z^{-1}(F(x_i))$$
$$F(x_i) = \int_0^{x_i} f(x, \beta, \alpha)\, dx \tag{6}$$

where $Z^{-1}$ means the inverse cumulative distribution function of the normal distribution, while $F(x)$ is the cumulative distribution function of the gamma distribution. Here, dry and wet extremes were defined as SSIs smaller than -1.28 (a probability of 10%) and larger than 1.28 respectively.

The relative changes in frequency of dry/wet extremes between the reference period and different warming periods were first calculated for each GCM under each SSP scenario, and the ensemble means were then determined for each warming level. To quantify the uncertainty, the above calculations were repeated by using the bootstrap 10,000 times, and 11 GCMs were resampled with replacement during each bootstrap (Christopher et al., 2018). The 5% and 95% percentiles of the total 10,000 estimations were finally taken as the 5~95% uncertainty ranges.

**3 Results**

**3.1 Terrestrial hydrological changes at different warming levels**

As shown in Figures 1b-1e, observations (pink lines) show that the annual temperature, precipitation and growing season LAI increase at the rates of 0.63°C/decade (p=0), 16.9 mm/decade (p=0.02), and 0.02 $m^2/m^2$/decade (p=0.001) during 1979-2014 respectively. The ensemble means of CMIP6 simulations (black lines) can generally capture the historical increasing trends of temperature (0.30 °C/decade, p=0), precipitation (7.1 mm/decade, p=0) and growing season LAI (0.029 $m^2/m^2$/decade, p=0), although the trends for precipitation and temperature are

underestimated. In 2015-2100, the SSP245 scenario (blue lines) shows continued
warming, wetting and greening trends, and the trends are larger in the SSP585
scenario (red lines). The $CO_2$ concentration also keeps increasing during 2015-2100
and reaches to 600 ppm and 1150 ppm in 2100 for the SSP245 and SSP585 scenarios
respectively. Although the SSP585 scenario reaches the same warming levels earlier
than the SSP245 scenario (Table 2), there is no significant difference between them in
the meteorological variables during the same warming period (not shown). Thus, we
do not distinguish SSP245 and SSP585 scenarios at the same warming level in the
following analysis.
Figure 3 and Table 3 show the evaluation of model simulation. Driven by
observed meteorological and ecological forcings, the CMFD/CSSPv2 simulates
monthly streamflow over the Yellow and Yangtze river headwaters quite well. The
Kling-Gupta efficiencies of CMFD/CSSPv2 simulated monthly streamflow are 0.94
and 0.91 over Tangnaihai (TNH) and Zhimenda (ZMD) stations, respectively. The
simulated monthly Terrestrial Water Storage Anomaly (TWSA) during 2003-2014 in
CMFD/CSSPv2 also agrees with the GRACE satellite observation and captures the
increasing trend. For the interannual variations of evapotranspiration, CMFD/CSSPv2
is consistent with the ensemble mean of the GLEAM_ET and MTE_ET products, and
the correlation coefficient and root mean squared error (RMSE) during 1982-2011 are
0.87 (p<0.01) and 14 mm/year respectively. This suggests the good performance of
the CSSPv2 in simulating the hydrological processes over the Sanjiangyuan region.
Although meteorological and ecological outputs from CMIP6 models have coarse
resolutions (~100 km), the land surface simulation driven by bias corrected CMIP6
results (CMIP6_His/CSSPv2) also captures the terrestrial hydrological variations
reasonably well. The Kling-Gupta efficiency of the ensemble mean streamflow
simulation reaches up to 0.71~0.81, and the ensemble mean monthly Terrestrial Water
Storage Anomaly (TWSA) and annual evapotranspiration generally agree with
observations and other reference data (Figures 3c-3d).
Figure 4 shows relative changes of terrestrial hydrological variables over the
Sanjiangyuan region at different warming levels. The ensemble mean of the increase
in annual precipitation is 5% at 1.5°C warming level, and additional 0.5°C and 1.5°C
warming will further increase the wetting trends to 7% and 13% respectively. Annual
evapotranspiration experiences significant increases at all warming levels, and the
ensemble mean increases are 4%, 7% and 13% at 1.5, 2.0 and 3.0°C warming levels
respectively. The ratio of transpiration to evapotranspiration also increases
significantly, indicating that vegetation transpiration increases much larger than the
soil evaporation and canopy evaporation. Although annual total runoff has larger
relative changes than evapotranspiration (6%, 9% and 14% at 1.5, 2.0 and 3.0°C
warming levels respectively), the uncertainty is large as only 75% of the models show
positive signals, which may be caused by large uncertain changes during summer and
autumn seasons. The terrestrial water storage (TWS) which includes foliage water,
surface water, soil moisture and groundwater, shows slightly decreasing trend at
annual scale, suggesting that the increasing precipitation in the future becomes extra
evapotranspiration and runoff instead of recharging the local water storage. The
accelerated terrestrial hydrological cycle also exists at seasonal scale, as the seasonal
changes are consistent with the annual ones.
**3.2 Changes in streamflow extremes at different warming levels**
Although the intensified terrestrial hydrology induces more streamflow over the
headwater region of Yellow river during winter and spring months, streamflow does
not increase and even decreases during the flood season (July-September; Figure 5a).
Figure 5b shows the changes of streamflow dry extremes over the Yellow river source
region at different warming levels, with the error bars showing estimated uncertainties.
The frequency of streamflow dry extremes over the Yellow river is found to increase
by 55% at 1.5°C warming level (Figure 5b), but the uncertainty is larger than the
ensemble mean. However, the dry extreme frequency will further increase to 77% and
125% at the 2.0 and 3.0°C warming levels and the results become significant (Figure
5b). No statistically significant changes are found for the wet extremes at all warming
levels over the Yellow River headwater region, as the uncertainty ranges are larger
than the ensemble means.
Over the Yangtze river headwater region, streamflow increases in all months at
different warming levels (Figure 5c). The frequency of wet extremes increases
significantly by 138%, 202% and 232% at 1.5, 2.0 and 3.0°C warming levels (Figure
5d), suggesting a higher risk of flooding. Although the frequency of dry extremes also
tends to decrease significantly by 35%, 44%, 34% at the three warming levels, the
changes are much smaller than those of the wet extremes. Moreover, contributions
from climate change and ecological change are both smaller than the uncertainty
ranges (not shown), suggesting that their impacts on the changes of dry extremes over
the Yangtze river headwater region are not distinguishable. Thus, we mainly focus on
the dry extremes over the Yellow river and the wet extremes over the Yangtze river in
the following analysis.
Different changes of streamflow extremes over the Yellow and Yangtze rivers
can be interpreted from different accelerating rates of precipitation and
evapotranspiration. Figure 6 shows probability density functions (PDFs) of
precipitation, evapotranspiration and their difference (P-ET, i.e. residual water for
runoff generation) during the flood season. Over the Yellow river, PDFs of
precipitation and evapotranspiration both shift to the right against the reference period,
except for the precipitation at 1.5°C warming level. However, the increasing trend of
evapotranspiration is stronger than that of precipitation, leading to a left shift for the
PDF of P-ET. Moreover, increased variations of precipitation and evapotranspiration,
as indicated by the increased spread of their PDFs, also lead to a larger spread of
PDFs of P-ET. The above two factors together induce a heavier left tail in the PDF of
P-ET for the warming future than the reference period (Figure 6e). The probability of
P-ET<80mm increases from 0.1 during historical period to 0.11, 0.13 and 0.16 at 1.5,
2.0 and 3.0°C warming levels individually. This indicates a higher probability of less
water left for runoff generation at different warming levels, given little changes in
TWS (section 3.1). Moreover, Figure 6e also shows little change in the right tails of
the PDF of P-ET as probability for P-ET>130mm stays around 0.1 at different
warming levels, suggesting little change to the probability of high residual water. This
is consistent with the insignificant wet extreme change over the Yellow river. Over the
Yangtze river, however, intensified precipitation is much larger than the increased
evapotranspiration, leading to a systematic rightward shift of the PDF of P-ET
(Figures 6b, 6d and 6f). Thus both the dry and wet extremes show significant changes
over the Yangtze river.

**3.3 Influences of land cover change and $CO_2$ physiological forcing**

Figures 7a-7b show the changes of streamflow extremes (compared with the
reference period) induced by climate and ecological factors. Although the contribution
from climate change (red bars in Figures 7a-7b) is greater than the ecological factors
(blue and cyan bars in in Figures 7a-7b), influences of $CO_2$ physiological forcing and
land cover change are nontrivial. The $CO_2$ physiological forcing tends to alleviate dry
extremes (or increase wet extremes), while land cover change plays a contrary role.
Over the Yellow river, the combined impact of the two ecological factors (sum of blue
and cyan bars) reduces the increasing trend of dry extremes caused by climate change
(red bars) by 18~22% at 1.5 and 2.0 °C warming levels, while intensifies the dry
extremes by 9% at 3.0°C warming level. This can be interpreted from their
contributions to the evapotranspiration, as enhancement effect of the increased LAI on
ET is weaker than the suppression effect of $CO_2$ physiological forcing at 1.5 and
2.0°C warming levels, while stronger at 3.0°C warming level (not shown). Over the
Yangtze river, similarly, combined effect of land cover and $CO_2$ physiological forcing
increases the wet extremes by 9% at 1.5°C warming level while decreases the wet
extremes by 12% at 3.0°C warming level.
In addition, Figures 7c and 7d show that the combined impact of $CO_2$
physiological forcing and land cover change also influences the differences between
different warming levels. Over the Yellow river, climate change increases dry
extremes by 26% from 1.5 to 2.0°C warming level, and by 40% from 1.5 and 3.0°C
warming level (red bars in Figure 7c). After considering the two ecological factors
(pink bars in Figure 7c), above two values change to 22% and 70% respectively, and
the difference between 1.5 and 3.0°C warming levels becomes significant. For the wet
extreme over the Yangtze river (Figure 7d), the climate change induced difference
between 1.5 and 2.0°C warming levels is decreased by 16% after accounting for the
two ecological factors. And this decrease reaches up to 49% for the difference
between 1.5 and 3.0°C warming levels. We also compared the scenarios when $CO_2$
physiological forcing and land cover change are combined with climate change
individually (blue and cyan bars in Figures 7c-d), and the results show the land cover
change dominates their combined influences on the difference between different
warming levels.
**4 Conclusions and Discussion**
This study investigates changes of streamflow extremes over the Sanjiangyuan
region at different global warming levels through high-resolution land surface
modeling driven by CMIP6 climate simulations. The terrestrial hydrological cycle
under global warming of 1.5°C is found to accelerate by 4~6% compared with the
reference period of 1985-2014, according to the relative changes of precipitation,
evapotranspiration and total runoff. The terrestrial water storage, however, shows a

slight but significant decreasing trend as increased evapotranspiration and runoff are larger than the increased precipitation. This decreasing trend of terrestrial water storage in the warming future is also found in six major basins in China (Jia et al., 2020). Although streamflow changes during the flood season has a large uncertainty, the frequency of wet extremes over the Yangtze river will increase significantly by 138% and that of dry extremes over the Yellow river will increase by 55% compared with that during 1985~2014. With an additional 0.5°C warming, the frequency of dry and wet extremes will increase further by 22~64%. If the global warming is not adequately managed (e.g., to reach 3.0°C), wet extremes over the Yangtze river and dry extremes over the Yellow river will increase by 232% and 125%. The changes from 1.5 to 2.0 and 3.0°C are nonlinear compared with that from reference period to 1.5°C, which are also found for some fixed-threshold climate indices over the Europe (Dosio and Fischer, 2018). It is necessary to cap the global warming at 2°C or even lower level, to reduce the risk of wet and dry extremes over the Yangtze and Yellow rivers.

This study also shows the nontrivial contributions from land cover change and $CO_2$ physiological forcing to the extreme streamflow changes especially at 2.0 and 3.0°C warming levels. The $CO_2$ physiological forcing is found to increase streamflow and reduce the dry extreme frequency by 14~24%, which is consistent with previous findings that $CO_2$ physiological forcing would increase available water and reduce water stress at the end of this century (Wiltshire et al., 2013). However, our results further show that the drying effect of increasing LAI on streamflow will exceed the

wetting effect of $CO_2$ physiological forcing at 3.0°C warming level (during
2048~2075) over the Sanjiangyuan region, making a reversion in the combined
impacts of $CO_2$ physiological forcing and land cover. Thus it is vital to consider the
impact of land cover change in the projection of future water stress especially at high
warming scenarios.
Moreover, about 43~52% of the extreme streamflow changes between 1.5 and
3.0°C warming levels are attributed to the increased LAI. Considering the LAI
projections from different CMIP6 models are induced by the climate change, it can be
inferred that the indirect influence of climate change (e.g., through land cover change)
has the same and even larger importance on the changes of streamflow extremes
between 1.5 and 3.0°C or even higher warming levels, compared with the direct
influence (e.g., through precipitation and evapotranspiration). Thus, it is vital to
investigate hydrological and its extremes changes among different warming levels
from an eco-hydrological perspective instead of focusing on climate change alone.
Although we used 11 CMIP6 models combined with two SSP scenarios to reduce
the uncertainty of future projections caused by GCMs, using a single land surface
model may result in uncertainties (Marx et al., 2018). However, considering the good
performance of the CSSPv2 land surface model over the Sanjiangyuan region and the
dominant role of GCMs' uncertainty (Zhao et al., 2019; Samaniego et al., 2017),
uncertainty from the CSSPv2 model should have limited influence on the robustness
of the result.

**Acknowledgments** We thank the World Climate Research Programme's Working
Group on Couple modelling for providing CMIP6 data (https://esgf-node.llnl.gov).
This work was supported by National Key R&D Program of China
(2018YFA0606002) and National Natural Science Foundation of China (41875105,
91547103), and the Startup Foundation for Introducing Talent of NUIST.

**Competing interests**
The authors declare that they have no conflict of interest.

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

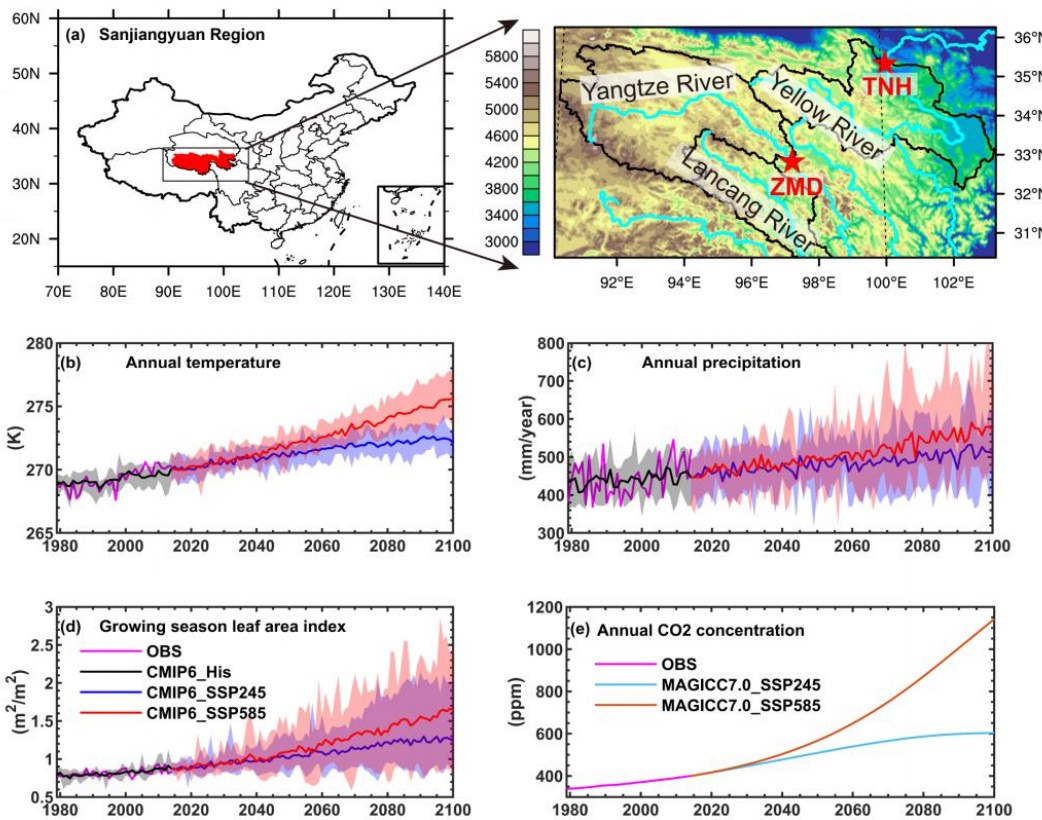

**Figure 1.** (a) The locations of the Sanjiangyuan region and streamflow gauges. (b)-(d)

The time series of annual temperature, precipitation, and growing season leaf area

index averaged over the Sanjiangyuan region during 1979-2100. (e) Observed and

simulated annual $CO_2$ concentration over the Sanjiangyuan region. Red pentagrams in

(a) are two streamflow stations named Tangnaihai (TNH) and Zhimenda (ZMD).

Black, blue and red lines in (b-d) are ensemble means of CMIP6 model simulations

from the historical, SSP245 and SSP585 experiments. Shadings are ranges of

individual ensemble members. Cyan and brown lines in (e) are future $CO_2$

concentration under SSP245 and SSP585 scenarios simulated by MAGICC7.0 model.

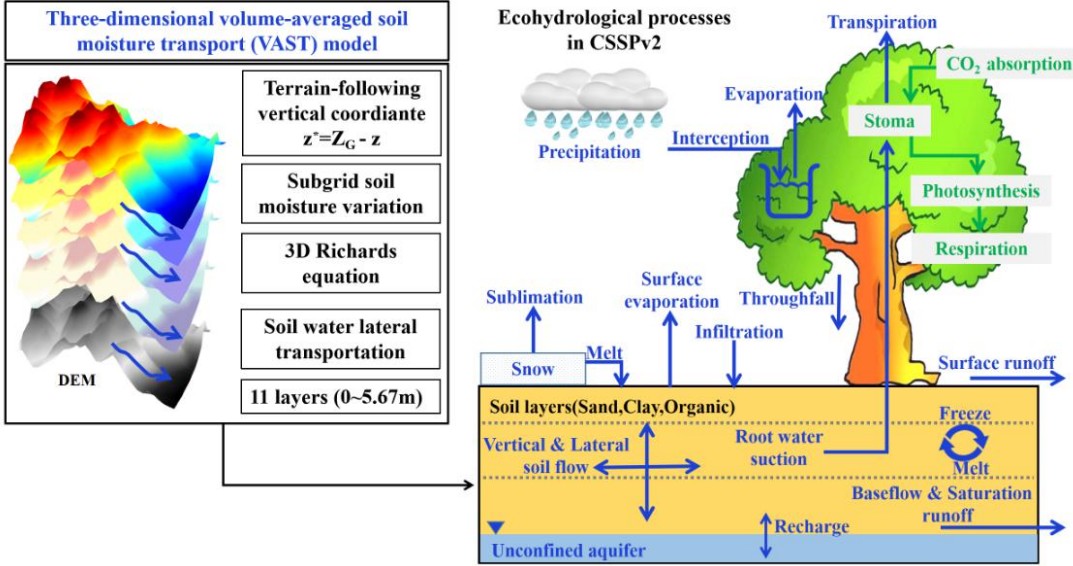

**Figure 2.** Main ecohydrological processes in the Conjunctive Surface-Subsurface

Process version 2 (CSSPv2) land surface model.

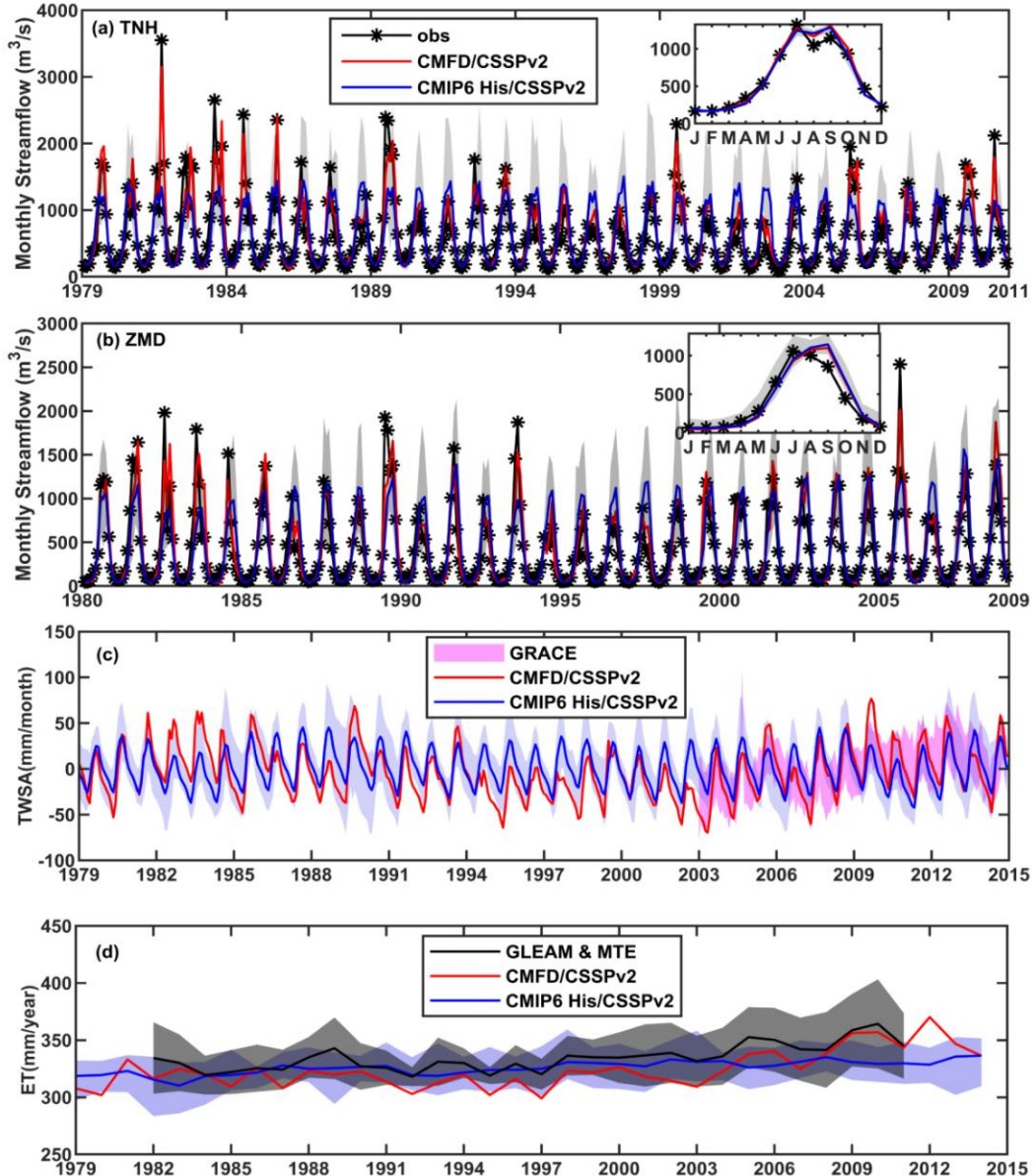

**Figure 3.** Evaluation of model simulations. (a-b) Observed and simulated monthly

streamflow at the Tangnaihai (TNH) and Zhimenda (ZMD) hydrological stations, with

the climatology shown in the upper-right corner. (c-d) Evaluation of the simulated

monthly terrestrial water storage anomaly (TWSA) and annual evapotranspiration (ET)

averaged over the Sanjiangyuan region. Red lines are CSSPv2 simulation forced by

observed meteorological forcing. Blue lines represent ensemble means of 11

CMIP6_His/CSSPv2 simulations, while gray shadings in (a-b) and blue shadings in

(c-d) are ranges of individual ensemble members. Pink shading in (c) is GRACE

satellite observations. Black line and black shading in (d) are ensemble mean and

ranges of GLEAM_ET and MTE_ET datasets.

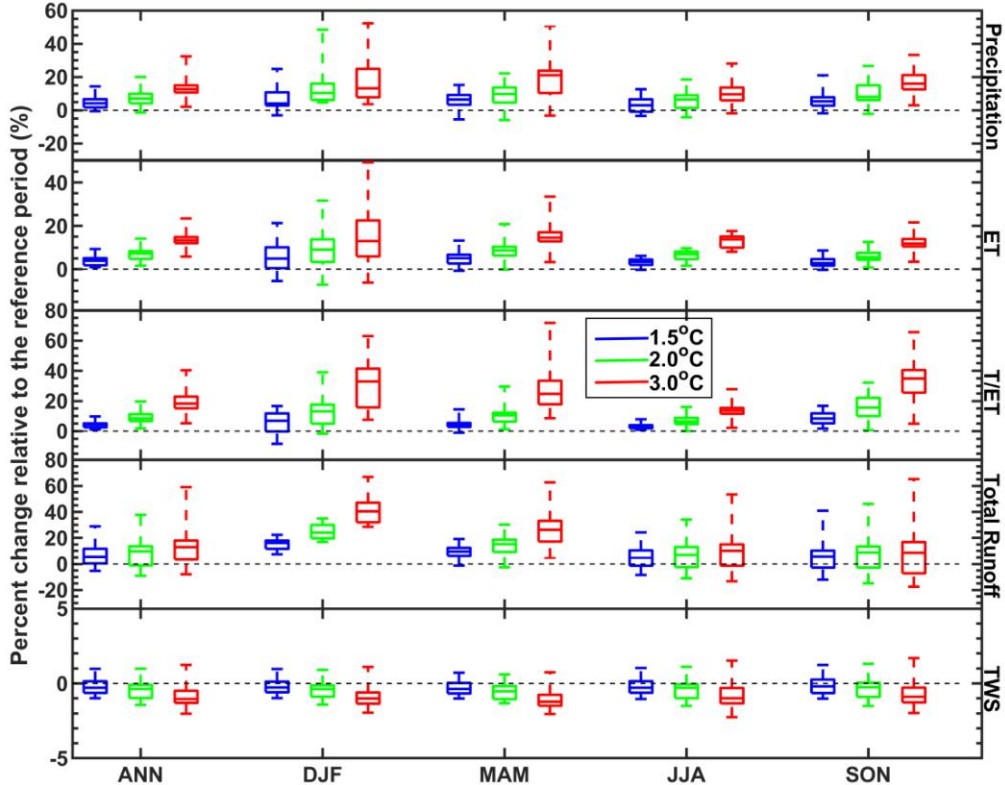

**Figure 4.** Box plots of relative changes of regional mean precipitation, evapotranspiration (ET), ratio of transpiration to evapotranspiration (T/ET), total runoff and terrestrial water storage (TWS) at different global warming levels. Reference period is 1985-2014, and annual (ANN) and seasonal (winter: DF, spring: MAM, summer: JJA and autumn: SON) results are all shown. Boxes show 25th to 75th ranges among 22 CMIP6_SSP/CSSPv2 simulations, while lines in the boxes are median values.

664

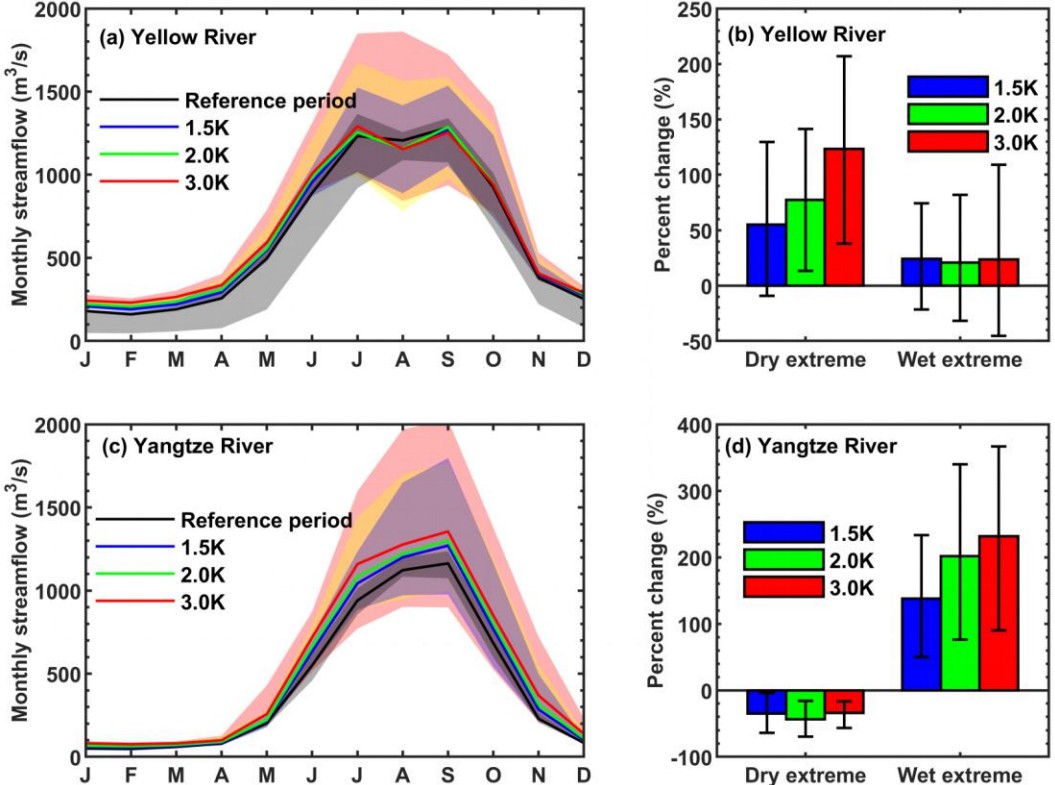

665

**Figure 5.** Changes of streamflow and its extremes at the outlets of the headwater

regions of the Yellow river and the Yangtze river, i.e., Tangnaihai gauge and

Zhimenda gauge. (a) Simulated monthly streamflow over the Yellow river during the

reference period (1985-2014) and the periods with different global warming levels.

Solid lines represent ensemble means, while shadings are ranges of individual

ensemble members. (b) Percent changes in frequency of dry and wet extremes in

July-September at different warming levels. Colored bars are ensemble means, while

error bars are 5~95% uncertainty ranges estimated by using bootstrapping for 10,000

times. (c) and (d) are the same as (a) and (b), but for the Yangtze river.

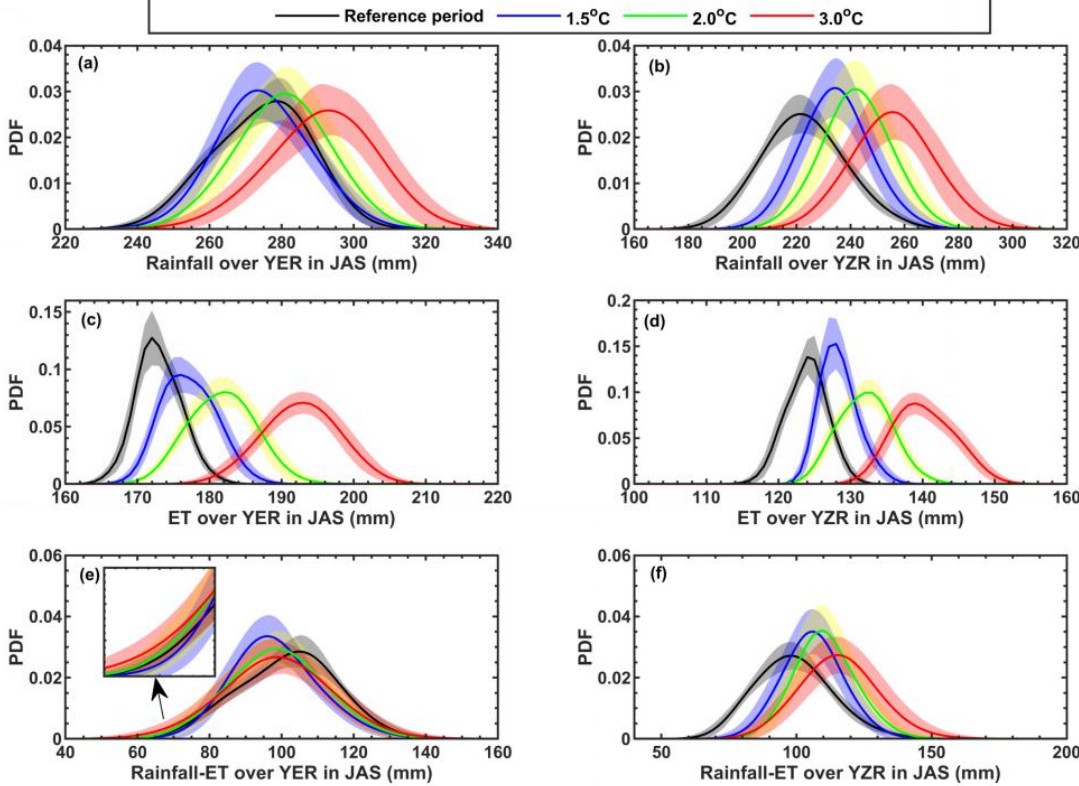

**Figure 6.** Probability density functions (PDFs) of regional mean rainfall, evapotranspiration (ET) and their difference over the headwater regions of Yellow river (YER) and Yangtze river (YZR) during flooding seasons (July-September) for the reference period (1985-2014) and the periods with 1.5, 2.0 and 3.0°C global warming levels. Shadings are 5~95% uncertainty ranges.

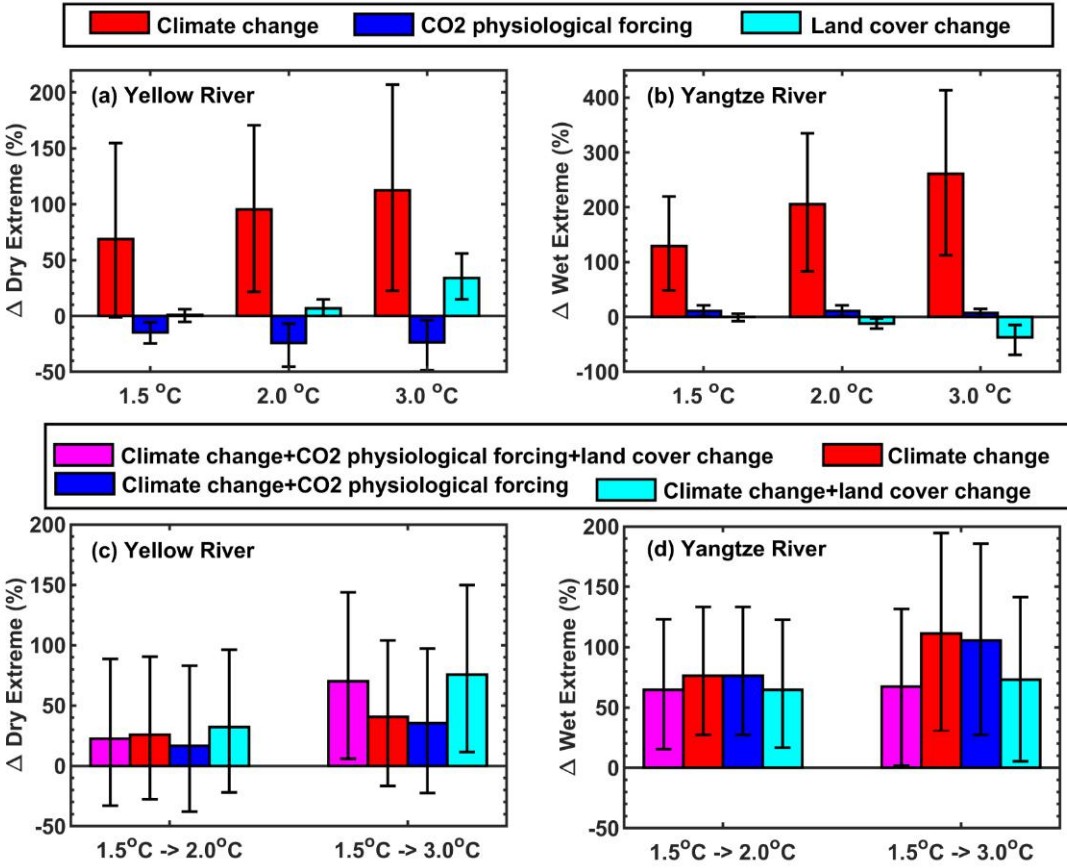

683

**Figure 7.** (a-b) Influences of climate change, $CO_2$ physiological forcing and land cover change on relative changes in frequency of the dry and wet extremes in July-September at different global warming levels for the headwater regions of Yellow river and Yangtze river. (c-d) Changes of dry and wet extremes under additional warming of 0.5°C and 1.5°C with the consideration of different factors. All the changes are relative to the reference period (1985-2014). Ensemble means are shown by colored bars while the 5~95% uncertainty ranges estimated by using bootstrapping for 10,000 times are represented by error bars.


**Table 1.** CMIP6 simulations used in this study. His means historical simulations
during 1979-2014 with both anthropogenic and natural forcings, SSP245 and SSP585
represent two Shared Socioeconomic Pathways during 2015-2100. Note the
CNRM-CM6-1 and CNRM-ESM2-1 do not provide r1i1p1f1 realization, so r1i1p1f2
was used instead.

| No. | Models | Experiments | Realization | Horizontal Resolution (Longitude × Latitude Grid Points) |
|---|---|---|---|---|
| 1 | ACCESS-ESM1-5 | His/SSP245/SSP585 | r1i1p1f1 | 192×145 |
| 2 | BCC-CSM2-MR | His/SSP245/SSP585 | r1i1p1f1 | 320×160 |
| 3 | CESM2 | His/SSP245/SSP585 | r1i1p1f1 | 288×192 |
| 4 | CNRM-CM6-1 | His/SSP245/SSP585 | r1i1p1f2 | 256×128 |
| 5 | CNRM-ESM2-1 | His/SSP245/SSP585 | r1i1p1f2 | 256×128 |
| 6 | EC-Earth3-Veg | His/SSP245/SSP585 | r1i1p1f1 | 512×256 |
| 7 | FGOALS-g3 | His/SSP245/SSP585 | r1i1p1f1 | 180×80 |
| 8 | GFDL-CM4 | His/SSP245/SSP585 | r1i1p1f1 | 288×180 |
| 9 | INM-CM5-0 | His/SSP245/SSP585 | r1i1p1f1 | 180×120 |
| 10 | MPI-ESM1-2-HR | His/SSP245/SSP585 | r1i1p1f1 | 384×192 |
| 11 | MRI-ESM2-0 | His/SSP245/SSP585 | r1i1p1f1 | 320×160 |


**Table 2.** Determination of "crossing years" for the periods reaching 1.5, 2 and 3°C
warming levels for different GCM and SSP combinations.

| Models | 1.5°C warming level | | 2.0°C warming level | | 3.0°C warming level | |
|--------|--------|--------|--------|--------|--------|--------|
| | SSP245 | SSP585 | SSP245 | SSP585 | SSP245 | SSP585 |
| ACCESS-ESM1-5 | 2024 | 2023 | 2037 | 2034 | 2070 | 2052 |
| BCC-CSM2-MR | 2026 | 2023 | 2043 | 2034 | Not found | 2054 |
| CESM2 | 2024 | 2022 | 2037 | 2032 | 2069 | 2048 |
| CNRM-CM6-1 | 2032 | 2028 | 2047 | 2039 | 2075 | 2055 |
| CNRM-ESM2-1 | 2030 | 2026 | 2049 | 2039 | 2075 | 2058 |
| EC-Earth3-Veg | 2028 | 2023 | 2044 | 2035 | 2072 | 2053 |
| FGOALS-g3 | 2033 | 2032 | 2063 | 2046 | Not found | 2069 |
| GFDL-CM4 | 2025 | 2024 | 2038 | 2036 | 2073 | 2053 |
| INM-CM5-0 | 2031 | 2027 | 2059 | 2038 | Not found | 2063 |
| MPI-ESM1-2-HR | 2032 | 2030 | 2055 | 2044 | Not found | 2066 |
| MRI-ESM2-0 | 2024 | 2021 | 2038 | 2030 | 2074 | 2051 |


**Table 3.** Performance for CSSPv2 model simulations driven by the observed
meteorological forcing (CMFD/CSSPv2) and the bias-corrected CMIP6 historical
simulations (CMIP6_His/CSSPv2). The metrics include correlation coefficient (CC),
root mean squared error (RMSE), and Kling-Gupta efficiency (KGE). The KGE is
only used to evaluate streamflow.

| Variables | Experiments | CC | RMSE | KGE |
|---|---|---|---|---|
| Monthly streamflow at TNH station | CMFD/CSSPv2 | 0.95 | 165 m$^3$/s | 0.94 |
| | CMIP6_His/CSSPv2 | 0.76 | 342 m$^3$/s | 0.71 |
| Monthly streamflow at ZMD station | CMFD/CSSPv2 | 0.93 | 169 m$^3$/s | 0.91 |
| | CMIP6_His/CSSPv2 | 0.82 | 257 m$^3$/s | 0.81 |
| Monthly terrestrial water storage anomaly over the Sanjiangyuan region | CMFD/CSSPv2 | 0.7 | 22 mm/month | - |
| | CMIP6_His/CSSPv2 | 0.4 | 24 mm/month | - |
| Annual evapotranspiration over the Sanjiangyuan region | CMFD/CSSPv2 | 0.87 | 14 mm/year | - |
| | CMIP6_His/CSSPv2 | 0.47 | 13 mm/year | - |

