# Peer review of "Accelerated hydrological cycle over the Sanjiangyuan region induces"

_Hydrology and Earth System Sciences, 2020_

## Referee Comment (RC1) · Anonymous Referee #1 · 13 Aug 2020

General Comments: The paper presents an analysis of the impacts of climate change and two ecological factors ($CO_2$ physiological forcing and land cover change) for the streamflow extremes of the Sanjiangyuan region. The methodology used and the conclusions drawn are sound, and the manuscript is well structured. However, some questions needed to be explained clearly and the English writing of this manuscript needs improvement.

Specific comments: Lin17 on page 2: '∼700' change to '700'

Lin40 on page 4: 'Global temperature has been increasing' change to 'Global temperature has increased'

Lin 61-62 on page 5: The statement by the authors that 'Thus, it is necessary to assess their combined impacts on the projection of streamflow extremes at different warming levels' is confusing. This sentence needs to be clarified with more evidence to prove the veracity of the statements. In addition, the entire paragraph can be rephrase.

Lin 63-66 on page 5: The reasoning behind the choice of the streamflow extremes over the Sanjiangyuan regions needs to be explained. If historical changes in climate and ecology have significantly altered the terrestrial hydrology over the regions, the terrestrial hydrology also need analysis. At the same time, the characteristics of basin of the headwaters of the Yellow river and Yangtze river should be provided, such as area and discharge.

Lin 67-72 on page 5: Does $CO_2$ physiological forcing has a significant influence on the terrestrial hydrology and its extremes in Sanjiangyuan or other high-land areas? It would be better to add some related literature.

Lin 91-94 on page 6: Streamflow observations are daily or monthly streamflow observations? It seem monthly streamflow in this work.

Lin 107-109 on page 7: In this study, 11 models in CMIP6 that can reproduce the increased precipitation over the Sanjiangyuan, were chosen for the analysis. Please give more explanation why only precipitation was considered. In addition, can those models correctly simulate the temperature, specific humidity, etc.?

Lin 143-148 on page 7: It is important to show the structure of the model and how it handles the various hydrological processes as mentioned in this part. Maybe you can insert a figure of the structure of the eco-hydrological model.

Lin 196-198 on page 11: 'the ensemble means of CMIP6 simulations can reproduce the historical increasing trends of temperature, precipitation, and LAI reasonably well.' As shown in the figure.1(d), the ensemble means of CMIP6 seem to hardly simulate the trend of the precipitation. Please give more explanations for this.

[Figure]

Lin 207-218 on page 11: In this paragraph, the author used different indices to measure the performance of models including ling-Gupta efficiencies, correlation coefficient, and root mean squared error (RMSE). A simple introduction of those indices can be added in section 2. In addition, the statistical results of the indices in this study can be presented in a table.

Lin 254-255 on page 13: No significant changes? As shown in Figure 4b, the frequency of wet extremes tends to increase by 25%. Please give more explanation.

Lin 261-264 on page 14: 'Moreover, the frequency of dry extremes tends to decrease significantly ..' It seem that the dry extremes over the Yangtze river also need further analysis at different global warming levels. Please clarify.

Lin 298-300 on page 15: Please clarify this sentence.

Lin 321-323 on page 11: A section on uncertainties should be included. Climate model and eco-hydrological model are sources of uncertainties. For example, according to Fig 2, the simulations tend to underestimate the high flow, which will inevitably affect the results.

Figure 1.(d) 'growthing season leaf area index' , while Line 483 'growing season leaf area index'?

Figure 4.(1) 'Simulated monthly streamflow climatology' change 'Simulated monthly streamflow'

---

## Referee Comment (RC2) · Anonymous Referee #2 · 28 Aug 2020

This is an interesting paper that analyzes the future changes in the streamflow extremes and its contributions from ecological factors over the Sanjiangyuan region based on observational data and model outputs driven by the CMIP6 data. Besides a regional accelerated hydrological cycle at different warming levels, the high risk of dry and wet extremes over the headwater of Yellow river and Yangtze river are also found. More importantly, the individual and combined impacts of land cover change and $CO_2$ physiological forcing on projected hydrological changes are figured out and emphasized. Overall, the manuscript is well structured and presented, and there are a few minor comments below.
1. Line 156: I suggest references for CLM and CoLM are required here.

2. Lines 242-243: How to understand the phenomena that both the ET and runoff increase with the increase in precipitation, while the local water storage TWS changes little? Is it a common issue in the accelerated hydrological cycle in other regions? Maybe a further explanation for the little TWS change would be useful.

3. Line 251: Is "55%" statistically significant? I also suggest the significance tests for the rest of the changes at different warming levels in sections 3.2 and 3.3.

4. Line 270: In Figure 5a, the PDF of precipitation at 1.5 degrees warming level doesn't shift to the right against the reference period. Please correct the statement.

5. Lines 269-272 and Lines 281-282: "Over the Yellow river… the increasing trend of ET is stronger than that of precipitation". "Over the Yangtze river, however, intensified ET is much smaller than the increased precipitation". How to understand the opposite phenomenon over the two regions? The change in ET significantly influences the streamflow extremes changes over the Yellow and Yangtze rivers headwaters. Maybe a brief mention of that here would be useful.

6. Line 278: Change "Figure 3e" to "Figure 5e"?

7. Lines 274-280: "The above two factors together induce a heavier left tail in the PDFs of P-ET for the warming future than the reference period (Figure 5e). This indicates a higher probability of less water left for runoff generation at different warming levels, given little changes in TWS (section 3.1). Moreover, Figure 3e also shows little change to the right tails in the PDF of P-ET (P-ET>130mm) at different warming levels, suggesting little change to the probability of high residual water." It's hard to clearly distinguish the "heavier left tail" and "little change to the right tails" in Figure 5e and thus explain the large dry extremes and insignificant wet extremes. Can you give a more clear clue for that?

8. Line 320: How to get the value of "4-6%" for the acceleration of the hydrological

cycle under global warming of 1.5 degrees?

9. Lines 323-324: What's the period for the change of streamflow extremesïij§

10. Lines 327-329: I'm not sure what does the "nonlinear changes" mean. Can you add some detail for the nonlinear changes from future warming over Europe?

11. Lines 347-350: "Considering the LAI projections from different CMIP6 models are induced by the climate change, it can be inferred that the indirect influence of climate change (e.g., through land cover change) has the same and even larger importance. . .compared with the direct influence (e.g., through precipitation and evapotranspiration)." How to understand the direct and indirect influence of climate change on the streamflow extremes changes? Can you give a further explanation for that?

---

## Author Response (AR1)

**Responses to the comments from Reviewer #1**

We are very grateful to the reviewer for the positive and careful review. The thoughtful comments have helped improve the manuscript. The reviewer's comments are italicized and our responses immediately follow.

*General Comments: The paper presents an analysis of the impacts of climate change and two ecological factors (CO2 physiological forcing and land cover change) for the streamflow extremes of the Sanjiangyuan region. The methodology used and the conclusions drawn are sound, and the manuscript is well structured. However, some questions needed to be explained clearly and the English writing of this manuscript needs improvement.*

**Response:** Thanks for the positive comments. We have made extensive modifications to our manuscript for clarification, and have proofread and edited the English carefully. Please see our responses below.

*Lin17 on page 2: '~700' change to '700'*
*Lin40 on page 4: 'Global temperature has been increasing' change to 'Global temperature has increased'*

**Response:** Revised as suggested. (L40 in the revised manuscript)

*Lin 61-62 on page 5: The statement by the authors that 'Thus, it is necessary to assess their combined impacts on the projection of streamflow extremes at different warming levels' is confusing. This sentence needs to be clarified with more evidence to prove the veracity of the statements. In addition, the entire paragraph can be rephrase.*

**Response:** Thanks for the suggestion. The necessity to assess the combined impacts of $CO_2$ physiological forcing and land cover change is due to their contrary influences on the terrestrial hydrology. We have rephrased the paragraph as follows:

"... In addition to climate change, recent works reveal the importance of the ecological factors (e.g., the $CO_2$ physiological forcing and land cover change) in modulating the streamflow and its extremes. For example, the increasing $CO_2$ concentration is found to alleviate the decreasing trend of future streamflow at global scale through decreasing the vegetation transpiration by reducing the stomatal conductance (known as the $CO_2$ physiological forcing) (Fowler et al., 2019; Wiltshire et al., 2013; Yang et al., 2019; Zhu et al., 2012). Contrary to the $CO_2$ physiological forcing, the vegetation greening in a warming climate is found to have a significant role in exacerbating hydrological drought, as it enhances transpiration and dries up the land (Yuan et al., 2018b). However, the relative contributions of $CO_2$ physiological forcing and vegetation greening to the changes in terrestrial hydrology especially the streamflow extremes are still unknown, and whether their combined impact changes at different warming levels needs to be investigated." (L51-71)

*Lin 63-66 on page 5: The reasoning behind the choice of the streamflow extremes over the Sanjiangyuan regions needs to be explained.*

**Response:** Thanks for the comment. The reason for the choice of the streamflow extremes over the Sanjiangyuan region is explained from two aspects in the revised manuscript as follows:

1) "Hosting the headwaters of the Yellow river, the Yangtze river and the Lancang-Mekong river, the Sanjiangyuan region is known as the "Asian Water Tower" and concerns 700 million people over its downstream areas. Changes in streamflow and its extremes over the Sanjiangyuan region not only influence the local ecosystems, environment and water resources, but also affect the security of food, energy, and water over the downstream areas." (L72-77)

2) "Both the regional climate and ecosystems show significant changes over the Sanjiangyuan region due to global warming (Bibi et al., 2018; Kuang and Jiao, 2016; Liang et al., 2013; Yang et al., 2013; Zhu et al., 2016), which makes it a sound region to investigate the role of climate change and ecological change (e.g., land cover change and CO2 physiological forcing) in influencing the streamflow and its extremes (Cuo et al., 2014; Ji and Yuan, 2018; Zhu et al., 2013)." (L77-83)

*If historical changes in climate and ecology have significantly altered the terrestrial hydrology over the regions, the terrestrial hydrology also need analysis.*

**Response:** Thanks for the comment. Actually, we have analyzed the terrestrial hydrological changes including the precipitation, evapotranspiration, total runoff and terrestrial water storage at different warming levels in section 3.1 and Figure 4. The results suggest that the regional hydrological cycle is accelerating in a warming climate. Please see the text in the last paragraph in section 3.1.

*At the same time, the characteristics of basin of the headwaters of the Yellow river*

*and Yangtze river should be provided, such as area and discharge.*

**Response:** Thanks for the suggestion. We have added a brief introduction to the characteristics of the study domain in the revised manuscript as follows:

"The Sanjiangyuan region is located at the eastern part of the Tibetan Plateau (Figure 1a), with the total area and mean elevation being $3.61 \times 10^5$ km$^2$ and 5000 m respectively. It plays a critical role in providing freshwater, by contributing 35%, 20% and 8% to the total annual streamflow of the Yellow, Yangtze and Lancang-Mekong rivers (Li et al., 2017; Liang et al., 2013). The source regions of Yellow, Yangtze and Lancang-Mekong rivers account for 46%, 44% and 10% of the total area of the Sanjiangyuan individually, and the Yellow river source region has a warmer climate and sparser snow cover than the Yangtze river source region." (L115-122)

*Lin 67-72 on page 5: Does CO2 physiological forcing has a significant influence on the terrestrial hydrology and its extremes in Sanjiangyuan or other high-land areas? It would be better to add some related literature.*

**Response:** Thanks for the suggestion. We have added some related literature as suggested: "And the $CO_2$ physiological forcing is revealed to cause equally large changes in regional flood extremes as the precipitation over the Yangtze and Mekong rivers (Fowler et al., 2019)." (L94-96)

Reference:

Fowler, M. D., Kooperman G. J., Randerson, J. T. and Pritchard M. S.: The effect of plant physiological responses to rising CO2 on global streamflow, Nat. Clim. Change, 9, 873-879, https://doi.org/10.1038/s41558-019-0602-x, 2019.

*Lin 91-94 on page 6: Streamflow observations are daily or monthly streamflow observations? It seem monthly streamflow in this work.*

**Response:** Yes, we used monthly streamflow to evaluate the model. We have clarified it as: "Monthly streamflow observations ..., were used to evaluate the streamflow simulations." (L123-125)

*Lin 107-109 on page 7: In this study, 11 models in CMIP6 that can reproduce the increased precipitation over the Sanjiangyuan, were chosen for the analysis. Please give more explanation why only precipitation was considered. In addition, can those models correctly simulate the temperature, specific humidity, etc.?*

**Response:** Thanks for the suggestion. We have evaluated the performance of CMIP6 models in representing the trends of other meteorological variables as suggested. Figure R1 shows that the ensemble mean (right panel) and each of the 11 CMIP6 model (left model) chosen in this research can reproduce the sign of historical trends of other meteorological forcings. We have revised the description to avoid misleading information: "... Then, models were chosen for the analysis when the simulated meteorological forcings (e.g., precipitation, temperature, humidity, and shortwave radiation) averaged over the Sanjiangyuan region have the same trend signs as the observations during 1979-2014. Table 1 shows the 11 CMIP6 models that were finally chosen in this study." (L139-145)

[Figure]

**Figure R1.** (a) Observed (purple line) and CMIP6 model simulated (black line) annual mean precipitation during 1979-2014. Shadings are ranges of all 11 CMIP6 models. The observed precipitation trends during 1979-2014 is shown by red circle on the right panel, while simulated trends of 11 CMIP6 models are shown by the boxplot. (b), (c) and (d) are the same as (a) but for temperature, humidity and shortwave radiation respectively.

*Lin 143-148 on page 7: It is important to show the structure of the model and how it handles the various hydrological processes as mentioned in this part. Maybe you can*

*insert a figure of the structure of the eco-hydrological model.*

**Response:** Thanks for the suggestion. Detailed model introduction and a new Figure 2 have been added in the revised paper as suggested:

"Figure 2 shows the structure and main ecohydrological processes in CSSPv2. The CSSPv2 is rooted in the Common Land Model (CoLM; Dai et al., 2003) with some improvements at hydrological processes. CSSPv2 has a volume-averaged soil moisture transport (VAST) model, which solves the quasi-three dimensional transportation of the soil water and explicitly considers the variability of moisture flux due to subgrid topographic variations (Choi et al., 2007). Moreover, the Variable Infiltration Capacity runoff scheme (Liang et al., 1994), and the influences of soil organic matters on soil hydrological properties were incorporated into the CSSPv2 by Yuan et al. (2018a), to improve its performance in simulating the terrestrial hydrology over the Sanjiangyuan region. Similar to CoLM and Community Land Model (Oleson et al., 2013), vegetation transpiration in CSSPv2 is based on Monin-Obukhov similarity theory, and the transpiration rate is constrained by leaf boundary layer and stomatal conductances. Parameterization of the stomatal conductance ($g_s$) in CSSPv2 is

$$g_s = m \frac{A_n}{P_{CO_2}/P_{atm}} h_s + b\beta_t$$

where the $m$ is a plant functional type dependent parameter, $A_n$ is leaf net photosynthesis ($\mu\, mol\, CO_2\, m^{-2}\, s^{-1}$), $P_{CO_2}$ is the $CO_2$ partial pressure at the leaf surface ($Pa$), $P_{atm}$ is the atmospheric pressure ($Pa$), $h_s$ is the lead surface humidity, $b$ is the minimum stomatal conductance ($\mu\, mol\, m^{-2}\, s^{-1}$), while $\beta_t$ is the soil water stress function. Generally, the stomatal conductance decreases with the increasing of $CO_2$ concentration. Generally, the stomatal conductance decreases with the increasing of $CO_2$ concentration." (L182-207)

[Figure]

**Figure 2.** Main ecohydrological processes in the Conjunctive Surface-Subsurface Process version 2 (CSSPv2) land surface model.

*Lin 196-198 on page 11: 'the ensemble means of CMIP6 simulations can reproduce the historical increasing trends of temperature, precipitation, and LAI reasonably well.' As shown in the figure.1(d), the ensemble means of CMIP6 seem to hardly simulate the trend of the precipitation. Please give more explanations for this.*

**Response:** Thanks for the suggestion. The previous statement " ... reproduce the historical increasing trends of ..." may cause some misunderstanding. We have revised it as: "As shown in Figures 1b-1e, observations (pink lines) show that the annual temperature, precipitation and growing season LAI increase at the rates of 0.63°C/decade (p=0), 16.9 mm/decade (p=0.02), and 0.02 $m^2/m^2$/decade (p=0.001) during 1979-2014 respectively. The ensemble means of CMIP6 simulations (black lines) can generally capture the historical increasing trends of temperature (0.30 °C/decade, p=0), precipitation (7.1 mm/decade, p=0) and growing season LAI (0.029 $m^2/m^2$/decade, p=0), although the trends for precipitation and temperature are underestimated." (L266-273)

*Lin 207-218 on page 11: In this paragraph, the author used different indices to measure the performance of models including ling-Gupta efficiencies, correlation coefficient, and root mean squared error (RMSE). A simple introduction of those indices can be added in section 2. In addition, the statistical results of the indices in this study can be presented in a table.*

**Response:** Thanks for the suggestion. We have added a brief description of the indices used in the research as: "Correlation coefficient (CC) and root mean squared error (RMSE) were calculated for validating the simulated monthly streamflow, annual evapotranspiration and monthly terrestrial water storage. The King-Gupta efficiency (KGE; Gupta et al., 2009), which is widely used in streamflow evaluations, was also calculated. Above metrics were calculated as follows:

$$CC = \frac{\sum_{i=1}^{n}(x_i - \bar{x})(y_i - \bar{y})}{\sqrt{\sum_{i=1}^{n}(x_i - \bar{x})^2 \sum_{i=1}^{n}(y_i - \bar{y})^2}}$$

$$RMSE = \sqrt{\frac{\sum_{i=1}^{n}(x_i - y_i)^2}{n}}$$

$$KGE = 1 - \sqrt{(1-CC)^2 + (1-\frac{\sigma_x}{\sigma_y})^2 + (1-\frac{\bar{x}}{\bar{y}})^2}$$

where $x_i$ and $y_i$ are observed and simulated variables in a specific month/year $i$ individually, and $\bar{x}$ and $\bar{y}$ are the corresponding monthly/annual means during the evaluation period $n$. The $\sigma_x$ and $\sigma_y$ are standard deviations for observed and simulated variables, respectively. The KGE ranges from negative infinity to 1, and model simulations can be regard as satisfactory when the KGE is larger than 0.5 (Moriasi et al., 2007)." (L211-224)

The statistical results of the indices are now shown in Table 3.

**Table 3.** Performance for CSSPv2 model simulations driven by the observed meteorological forcing (CMFD/CSSPv2) and the bias-corrected CMIP6 historical simulations (CMIP6_His/CSSPv2). The metrics include correlation coefficient (CC), root mean squared error (RMSE), and Kling-Gupta efficiency (KGE). The KGE is only used to evaluate streamflow.

| Variables | Experiments | CC | RMSE | KGE |
|---|---|---|---|---|
| Monthly streamflow at TNH station | CMFD/CSSPv2 | 0.95 | 165 m$^3$/s | 0.94 |
| | CMIP6_His/CSSPv2 | 0.76 | 342 m$^3$/s | 0.71 |
| Monthly streamflow at ZMD | CMFD/CSSPv2 | 0.93 | 169 m$^3$/s | 0.91 |

| station | CMIP6_His/CSSPv2 | 0.82 | 257 m³/s | 0.81 |
| Monthly terrestrial water storage anomaly over the Sanjiangyuan region | CMFD/CSSPv2 | 0.7 | 22 mm/month | - |
| | CMIP6_His/CSSPv2 | 0.4 | 24 mm/month | - |
| Annual evapotranspiration over the Sanjiangyuan region | CMFD/CSSPv2 | 0.87 | 14 mm/year | - |
| | CMIP6_His/CSSPv2 | 0.47 | 13 mm/year | - |

*Lin 254-255 on page 13: No significant changes? As shown in Figure 4b, the frequency of wet extremes tends to increase by 25%. Please give more explanation.*

**Response:** Thanks for the comment. We have clarified this as: "No statistically significant changes are found ..., as the uncertainty ranges are larger than the ensemble means." (L332-334)

*Lin 261-264 on page 14: 'Moreover, the frequency of dry extremes tends to decrease significantly ..' It seem that the dry extremes over the Yangtze river also need further analysis at different global warming levels. P2lease clarify.*

**Response:** Thanks for the suggestion. Although the dry extremes over the Yangtze river source region decrease significantly, contributions from the climate change and ecological factors cannot be distinguished due to the small changing magnitude. We have clarified it as:

"Although the frequency of dry extremes also tends to decrease significantly by 35%, 44%, 34% at the three warming levels, the changes are much smaller than those of the wet extremes. Moreover, contributions from climate change and ecological change are both smaller than the uncertainty ranges (not shown), suggesting that their impacts on the changes of dry extremes over the Yangtze river headwater region are not distinguishable. Thus, we mainly focus on the dry extremes over the Yellow river and the wet extremes over the Yangtze river in the following analysis." (L338-345)

*Lin 298-300 on page 15: Please clarify this sentence.*

**Response:** Thanks for the suggestion. We have revised this sentence and moved it to the front of detailed description of the importance of $CO_2$ physiological forcing and land cover change:

"Although the contribution from climate change (red bars in Figures 7a-7b) is greater than the ecological factors (blue and cyan bars in in Figures 7a-7b), influences of $CO_2$

physiological forcing and land cover change are nontrivial. ... Over the Yellow river, the combined impact of the two ecological factors ... reduces the increasing trend of dry extremes caused by climate change (red bars) by 18~22% at 1.5 and 2.0 °C warming levels, while intensifies the dry extremes by 9% at 3.0°C warming level. ... Over the Yangtze river, ... increases the wet extremes by 9% at 1.5°C warming level while decreases the wet extremes by 12% at 3.0°C warming level." (L371-385)

*Lin 321-323 on page 11: A section on uncertainties should be included. Climate model and eco-hydrological model are sources of uncertainties. For example, according to Fig 2, the simulations tend to underestimate the high flow, which will inevitably affect the results.*

**Response:** Thanks for the suggestion. We do agree with the reviewer that both global climate models (GCMs) and hydrological models are sources of uncertainties. Actually, we have used the bootstrap method to estimate the uncertainty caused by GCMs. We have added detailed information for the uncertainty estimations as follows:

"The relative changes in frequency of dry/wet extremes between the reference period and different warming periods were first calculated for each GCM under each SSP scenario, and the ensemble means were then determined for each warming level. To quantify the uncertainty, the above calculations were repeated by using the bootstrap 10,000 times, and 11 GCMs were resampled with replacement during each bootstrap (Christopher et al., 2018). The 5% and 95% percentiles of the total 10,000 estimations were finally taken as the 5~95% uncertainty ranges." (L257-263)

We do not add a new section to discuss the uncertainties, because analysis of uncertainties that caused by GCMs is already included in the results and only the robust changes are taken into consideration in this research.

"However, the dry extreme frequency will further increase to 77% and 125% at the 2.0 and 3.0°C warming levels and the results become significant (Figure 5b)."(L329-332).

"No statistically significant changes are found for the wet extremes at all warming levels over the Yellow River headwater region, as the uncertainty ranges are larger than the ensemble means."(L332-334)

However, we have added some discussions on the uncertainties caused by land surface hydrological model, as only one land surface model was used in this work. "Although we used 11 CMIP6 models combined with two SSP scenarios to reduce the uncertainty of future projections caused by GCMs, using a single land surface model may result in uncertainties (Marx et al., 2018). However, considering the good performance of the CSSPv2 land surface model over the Sanjiangyuan region and the dominant role of GCMs' uncertainty (Zhao et al., 2019; Samaniego et al., 2017), uncertainty from the CSSPv2 model should have limited influence on the robust of the result." (L447-453)

*Figure 1.(d) 'growthing season leaf area index' , while Line 483 'growing season leaf area index'?*

**Response:** We have corrected the 'growthing season leaf area index' as 'growing season leaf area index' in the revised Figure 1.

*Figure 4.(1) 'Simulated monthly streamflow climatology' change 'Simulated monthly streamflow'*

**Response:** Revised as suggested.

**Responses to the comments from Reviewer #2**

We are very grateful to the reviewer for the positive and careful review. The thoughtful comments have helped improve the manuscript. The reviewer's comments are italicized and our responses immediately follow.

*This is an interesting paper that analyzes the future changes in the streamflow extremes and its contributions from ecological factors over the Sanjiangyuan region based on observational data and model outputs driven by the CMIP6 data. Besides a regional accelerated hydrological cycle at different warming levels, the high risk of dry and wet extremes over the headwater of Yellow river and Yangtze river are also found. More importantly, the individual and combined impacts of land cover change and CO2 physiological forcing on projected hydrological changes are figured out and emphasized. Overall, the manuscript is well structured and presented, and there are a few minor comments below.*

**Response:** Thanks for the comment.

1. *Line 156: I suggest references for CLM and CoLM are required here.*

**Response:** Revised as suggested.

2. *Lines 242-243: How to understand the phenomena that both the ET and runoff increase with the increase in precipitation, while the local water storage TWS changes little? Is it a common issue in the accelerated hydrological cycle in other regions? Maybe a further explanation for the little TWS change would be useful.*

**Response:** Thanks for the suggestion. We have explained it as follows:

"The terrestrial water storage, however, shows a slight but significant decreasing trend as increased evapotranspiration and runoff are larger than the increased precipitation. This decreasing trend of terrestrial water storage in the warming future is also found in six major basins in China (Jia et al., 2020)." (L410-414)

3. *Line 251: Is "55%" statistically significant? I also suggest the significance tests for the rest of the changes at different warming levels in sections 3.2 and 3.3.*

**Response:** Thanks for the suggestion. We have clarified it as:

"The frequency of streamflow dry extremes over the Yellow river is found to increase by 55% at 1.5°C warming level (Figure 5b), but the uncertainty is larger than the ensemble mean." (L327-329)

Actually, we used the bootstrap method to estimate the uncertainty, and changes are considered to be significant when the ensemble mean is larger than the uncertainty range. We have clarified as:

"The relative changes in frequency of dry/wet extremes between the reference period and different warming periods were first calculated for each GCM under each SSP scenario, and the ensemble means were then determined for each warming level. To quantify the uncertainty, the above calculations were repeated by using the bootstrap 10,000 times, and 11 GCMs were resampled with replacement during each bootstrap (Christopher et al., 2018). The 5% and 95% percentiles of the total 10,000 estimations were finally taken as the 5~95% uncertainty ranges." (L257-263)

We have also added some statements on the uncertainties or significance in sections 3.2 and 3.3 as suggested:

"... the results become significant (Figure 5b). No statistically significant changes are found ..., as the uncertainty ranges are larger than the ensemble means."(L332-334)
"Moreover, contributions from climate change and ecological change are both smaller than the uncertainty ranges (not shown), suggesting that their impacts on the changes of dry extremes over the Yangtze river headwater region are not distinguishable." (L340-343)

*4. Line 270: In Figure 5a, the PDF of precipitation at 1.5 degrees warming level doesn't shift to the right against the reference period. Please correct the statement.*
**Response:** We have revised it as: "Over the Yellow river, PDFs of precipitation and evapotranspiration both shift to the right against the reference period, except for the precipitation at 1.5°C warming level." (L350-352)

*5. Lines 269-272 and Lines 281-282: "Over the Yellow river. . . the increasing trend of ET is stronger than that of precipitation". "Over the Yangtze river, however, intensified ET is much smaller than the increased precipitation". How to understand the opposite phenomenon over the two regions? The change in ET significantly influences the streamflow extremes changes over the Yellow and Yangtze rivers headwaters. Maybe a brief mention of that here would be useful.*

**Response:** Thanks for the comment. Actually, differences between headwaters of Yellow and Yangtze rivers are mainly caused by precipitation changes, as the increasing rate of ET at the Yangtze river headwater are similar to that at the Yellow river headwater. We have revised the statement as: "Over the Yangtze river, however, intensified precipitation is much larger than the increased evapotranspiration,...". (L365-366)

Different changing rates of precipitation over these two river source regions are beyond this work, so we do not discuss this in detail. Further work is needed to investigate the changes in horizontal moisture transport and local land-atmospheric exchanges.

*6. Line 278: Change "Figure 3e" to "Figure 5e"?*
**Response:** Thanks. We have changed "Figure 3e" to "Figure 6e" as a new figure was added to show the model structure.

*7. Lines 274-280: "The above two factors together induce a heavier left tail in the PDFs of P-ET for the warming future than the reference period (Figure 5e). This indicates a higher probability of less water left for runoff generation at different warming levels, given little changes in TWS (section 3.1). Moreover, Figure 3e also shows little change to the right tails in the PDF of P-ET (P-ET>130mm) at different warming levels, suggesting little change to the probability of high residual water." It's hard to clearly distinguish the "heavier left tail" and "little change to the right tails" in Figure 5e and thus explain the large dry extremes and insignificant wet extremes. Can you give a more clear clue for that?*
**Response:** Thanks for the suggestion. We have calculated the cumulative probability for both low and high P-ET values and added them in the manuscript to show the changes of PDFs more clearly.

"... together induce a heavier left tail in the PDF of P-ET .... The probability of P-ET<80mm increases from 0.1 during historical period to 0.11, 0.13 and 0.16 at 1.5, 2.0 and 3.0°C warming levels individually. ... shows little change to the right tails in the PDF of P-ET as probability for P-ET>130mm stays around 0.1 at different warming levels ..." (L357-362)

*8. Line 320: How to get the value of "4-6%" for the acceleration of the hydrological*

*cycle under global warming of 1.5 degrees?*

**Response:** Thanks for the comment. We have clarified as: "... is found to accelerate by 4~6% ..., according to the relative changes of precipitation, evapotranspiration and total runoff." (L409-410)

*9. Lines 323-324: What's the period for the change of streamflow extremes?*

**Response:** We have clarified as "Although ... compared with that during 1985~2014." (L416-417)

*10. Lines 327-329: I'm not sure what does the "nonlinear changes" mean. Can you add some detail for the nonlinear changes from future warming over Europe?*

**Response:** We have clarified the nonlinear changes as "The changes from 1.5 to 2.0 and 3.0°C are nonlinear compared with that from reference period to 1.5°C, ..." (L420-422)

To be specific, the wet extremes over Yangtze river source region increase by 138% at 1.5°C warming levels, which indicates a linear rate of 46%/0.5°C. However, projected change of wet extremes from 1.5 to 2.0°C warming levels is 64% which is much larger than the linear rate.

*11. Lines 347-350: "Considering the LAI projections from different CMIP6 models are induced by the climate change, it can be inferred that the indirect influence of climate change (e.g., through land cover change) has the same and even larger importance. . .compared with the direct influence (e.g., through precipitation and evapotranspiration)." How to understand the direct and indirect influence of climate change on the streamflow extremes changes? Can you give a further explanation for that?*

**Response:** The indirect influence of climate change means the climate change will induce land cover change and then the land cover change can also influence the hydrological extremes. The direct influence of climate change means the influence of meteorological forcings (e.g., precipitation, temperature, radiation) changes.

[revised manuscript text omitted]

---

## Editor Decision (ED1)

**Accelerated hydrological cycle over the Sanjiangyuan region induces**

**more streamflow extremes at different global warming levels**

          Peng Ji[1,2], Xing Yuan[3], Feng Ma[3], Ming Pan[4]

[1]Key Laboratory of Regional Climate-Environment for Temperate East Asia, Institute of Atmospheric Physics, Chinese Academy of Sciences, Beijing 100029, China

[2]College of Earth and Planetary Sciences, University of Chinese Academy of Sciences,

Beijing 1000493, China

[3]School of Hydrology and Water Resources, Nanjing University of Information

Science and Technology, Nanjing 210044, China

[4]Department of Civil and Environmental Engineering, Princeton University, Princeton,

New Jersey, USA

*Correspondence to*: Xing Yuan (xyuan@nuist.edu.cn)

**I suggest the authors to double check details of the manuscript and improve the language. I made some comments along my reading, which did absolutely not covering all of them.**

**Abstract.** Serving source water for the Yellow, Yangtze and Lancang-Mekong rivers, the Sanjiangyuan region concerns 700 million people over its downstream areas. Recent research suggests that the Sanjiangyuan region will become wetter in a warming future, but future changes in streamflow extremes remain unclear due to the complex hydrological processes over high-land areas and limited knowledge of the influences of land cover change and $CO_2$ physiological forcing. Based on high resolution land surface modeling during 1979~2100 driven by the climate and ecological projections from 11 newly released Coupled Model Intercomparison Project Phase 6 (CMIP6) climate models, we show that different accelerating rates of precipitation and evapotranspiration at 1.5 ℃ global warming level induce 55% more dry extremes over Yellow river and 138% more wet extremes over Yangtze river headwaters compared with the reference period (1985~2014). An additional 0.5 ℃ warming leads to a further nonlinear and more significant increase for both dry extremes over Yellow river (22%) and wet extremes over Yangtze river (64%). The combined role of $CO_2$ physiological forcing and vegetation greening, which used to be neglected in hydrological projections, is found to alleviate dry extremes at 1.5 and 2.0 ℃ warming levels but to intensify dry extremes at 3.0 ℃ warming level. Moreover, vegetation greening contributes half of the differences between 1.5 and 3.0 ℃ warming levels. This study emphasizes the importance of ecological processes in determining future changes in streamflow extremes, and suggests a "dry gets drier, wet gets wetter" condition over headwaters.

**Keywords** Terrestrial hydrological cycle, streamflow extremes, global warming levels,

CMIP6, Sanjiangyuan, land cover change

**1 Introduction**

**check the number**

Global temperature has increased at a rate of 1.7 °C/decade since 1970, contrary to the cooling trend over the past 8000 years (Marcott et al., 2013). The temperature measurements suggest that 2015-2019 is the warmest five years and 2010-2019 is also the warmest decade since 1850 (WMO, 2020). To mitigate the impact of this unprecedented warming on the global environment and human society, 195 nations adopted the Paris Agreement which decides to "hold the increase in the global average temperature to well below 2 °C above pre-industrial levels and pursing efforts to limit the temperature increase to 1.5 °C".

**are these references research papers or review articles? Suggest to provide several most recent review papers.**

[revised manuscript text omitted]

---

## Author Response (AR2)

Dear Prof. Tian,

Thank you for your decision letter on our manuscript entitled "Accelerated hydrological cycle over the Sanjiangyuan region induces more streamflow extremes at different global warming levels". We have carefully considered your suggestion, and proofread and edited the English. We hope that you find the revised manuscript and the response acceptable to *HESS*. Your comments are italicized and our responses immediately follow.

We appreciate the effort you spent to process the manuscript and look forward to hearing from you soon.

Sincerely yours,
Xing Yuan

*I suggest the authors to double check details of the manuscript and improve the language. I made some comments along my reading, which did absolutely not covering all of them.*

**Response:** Thanks for the suggestion. We have double checked the manuscript and edited the English carefully, including making the statement clearer, adding the equation numbers on the right-hand side, changing the tense and revising the grammar mistakes. Detailed information is shown below:

1. Change "... changes in streamflow extremes ..." to "... changes of streamflow extremes ..." L19

2. Change "The response of regional and global terrestrial hydrological processes, including streamflow and its extremes, to ..." to "The response of regional and global terrestrial hydrological processes (e.g., streamflow and its extremes) to ..." L48-50

3. Change "... because the increased $CO_2$ concentration will decrease the vegetation transpiration by reducing the stomatal conductance" to "... through decreasing the stomatal conductance and vegetation transpiration" L57-58

4. Change "... have a significant role in ..." to "... play a significant role in ..." L61

5. Change "... whether their combined impact changes at different warming levels ..." to "... whether their combined impact differs among different warming levels ..." L65-66

6. Change "Changes in streamflow and its extremes ... not only influence the local ecosystems..., but also affect the security of food..." to "Changes of streamflow and its extremes ... not only influence the local ecosystems ..., but also the security of food..." L69-71

7. Change "This makes it difficult to assess the ... on this vital headwaters region" to "Solving the above issues is essential for assessing the ... on this vital headwaters region." L95-96

8. Change "In this study, we investigate the ..." to "In this study, we investigated the ..." L97

9. Change "The combined impacts of ... are also quantified" to "The combined impacts of ... were also quantified" L100

10. Change "Monthly terrestrial water storage change observation and its uncertainty ... was provided ..." to "Estimations of monthly terrestrial water storage change and its uncertainty ... were provided ..." L118-120

11. Change "… were also used to evaluate the model performance on ET simulation." to "… were used to evaluate the ET simulation." L125-126

12. Change " (CMFD) is taken as meteorological observation" to "(CMFD) was taken as meteorological observation" L150

13. Change "the influences of soil organic matters on soil hydrological properties were incorporated into ..." to "... the hydrological properties of soil organic matters were incorporated into ..." L177-178

14. Add equation numbers. L185, 201-203, 240, 246

15. Change "were calculated for the observed and simulated ..., to evaluate the model performance." to "were calculated for validating the simulated ..." L196-198

16. Change "... are observed and simulated standard deviations respectively" to "... are standard deviations for observed and simulated variables respectively" L206-207

17. Change "Compared with the observation at Tangnaihai (TNH) and Zhimenda (ZMD) stations, the Kling-Gupta efficiencies of the CMFD/CSSPv2 simulated monthly streamflow are 0.94 and 0.91 respectively." to "The Kling-Gupta efficiencies of CMFD/CSSPv2 simulated monthly streamflow are 0.94 and 0.91 over Tangnaihai (TNH) and Zhimenda (ZMD) stations, respectively." L279-281

18. Change "by large uncertainty in the changes during summer and autumn seasons." to "by large uncertain changes during summer and autumn seasons." L308-309

19. Change "a left shift of PDF for P-ET." to "... a left shift for the PDF of P-ET."

L346-347

20. Change "shows little change to the right tails in the PDF" to "shows little change in the right tails of the PDF" L355

21. Change "as the increased LAI enhancement on ET is weaker than the suppression effect of $CO_2$ physiological forcing" to "as enhancement effect of the increased LAI on ET is weaker than the suppression effect of $CO_2$ physiological forcing" L373-375

22. Change "the robust of the result" to "the robustness of the result" L442

*L40, check the number*

**Response:** Thanks for the suggestion. We have revised as "0.17°C/decade" and double checked the number according to the reference. Moreover, other numbers are also rechecked.

*L50-51, are these references research papers or review articles? Suggest to provide several most recent review papers.*

**Response:** We have added the IPCC special report on the 1.5°C warming that reviewed pervious works comprehensively (Hoegh-Guldberg et al., 2018), as well as a recent paper (Xu et al., 2019) to the reference list. We still keep some representative research papers. The references are now: "(Döll et al., 2018; Hoegh-Guldberg et al., 2018; Marx et al., 2018; Mohammed et al., 2017; Thober et al., 2018; Xu et al., 2019; Zhang et al., 2016)" L50-52

*L77, where is this example?*

**Response:** The example is the work that investigated the influences of climate change and ecological change on the streamflow. We have revised the statement to avoid misunderstanding: "... Historical changes of climate and ecology (e.g., land cover) are found to cause significant reduction in mean and high flows over the Yellow River headwaters during 1979-2005, which potentially increases drought risk over its downstream areas (Ji and Yuan, 2018). And the $CO_2$ physiological forcing is revealed to cause equally large changes in regional flood extremes as the precipitation over the Yangtze and Mekong rivers (Fowler et al., 2019). Thus the Sanjiangyuan region is a sound region to investigate the role of climate change and ecological change ..." L75-83

*L228, need an equation to define SSI, which is important for understanding*

**Response:** Thanks for the suggestion. We have added equations and described the calculation of SSI in details.

"In this research, the standardized streamflow index (SSI) was used to define dry and wet extremes (Vicente-Serrano et al., 2012; Yuan et al., 2017). The July-August-September (JAS) mean streamflow for each year of the reference period was first collected and used to fit a gamma distribution:

$$f(x, \beta, \alpha) = \frac{\beta^\alpha}{\Gamma(\alpha)} x^{\alpha-1} e^{-\beta x} \tag{5}$$

where $x$ means streamflow while $\alpha$ and $\beta$ are parameters. Then the fitted distribution was used to standardize the JAS mean streamflow in each year ($i$) during both the reference and projection periods as:

$$SSI_i = Z^{-1}(F(x_i))$$
$$F(x_i) = \int_0^{x_i} f(x, \beta, \alpha)\, dx \tag{6}$$

where $Z^{-1}$ means the inverse cumulative distribution function of the normal distribution, while $F(x)$ is the cumulative distribution function of the gamma distribution. Here, dry and wet extremes were defined as SSIs smaller than -1.28 (a probability of 10%) and larger than 1.28 respectively." L236-250

*L428, change "Robust" to "Robustness"*
**Response:** Done as suggested.

*Figure 1, legend looks different from the Figure lines(e) says CO2 in north hemisphere, while text says in Sanjiangyuan. Check it.*
**Response:** Thanks for the suggestion. In this study, we did not use gridded CO2 concentration data due to large uncertainty (and very limited heterogeneity) at regional scale. Instead, we used the CO2 concentration averaged over the North Hemisphere to force the offline simulation over the Sanjiangyuan region (i.e., the CO2 concentrations are the same for each grid cell), which is also widely used in many impact studies. We have revised the figure and its caption as follows:

[revised manuscript text omitted]